# An Optical Tweezers-Based Single-Cell Manipulation and Detection Platform for Probing Real-Time Cancer Cell Chemotaxis and Response to Tyrosine Kinase Inhibitor PD153035

**Pei-Wen Peng** [1,†]**, Jen-Chang Yang** [2,3,4,†] **, Mamadi M.S Colley** [5] **and Tzu-Sen Yang** [1,3,4,5,*]

1   School of Dental Technology, Taipei Medical University, Taipei 110, Taiwan; apon@tmu.edu.tw
2   Graduate Institute of Nanomedicine and Medical Engineering, Taipei Medical University, Taipei 11031, Taiwan; yang820065@tmu.edu.tw
3   International PhD Program in Biomedical Engineering, Taipei Medical University, Taipei 110, Taiwan
4   Research Center of Biomedical Device, Taipei Medical University, Taipei 110, Taiwan
5   Graduate Institute of Biomedical Optomechatronics, Taipei Medical University, Taipei 110, Taiwan; ms_colley81@hotmail.com
*   Correspondence: tsyang@tmu.edu.tw; Tel.: +886-2-27-361-661 (ext. 5206)
†   P.-W.P. and J.-C.Y. contributed equally to this work as co-1st authors.

**Abstract:** We presented an approach to address cancer cell chemotaxis and response to tyrosine kinase inhibitor PD153035 at the single-cell level. We applied an optical tweezer system together with the platform at the single-cell level to manipulate an epidermal growth factor (EGF)-coated bead positioned close to the filopodia to locally stimulate HT29 cells, the human colon cancer cell line overexpressing the EGF receptor (EGFR). To address cancer cell chemotaxis, a single-cell movement model was also proposed to quantify the propagation speed at the leading and trailing edges of the cell along the chemosensing axis. This study focused on three perspectives: probing the chemosensing process mediated by EGF/EGFR signaling, investigating the mode of locomotion during the EGF-coated bead stimulation, and quantifying the effect of PD153035 on the EGF–EGFR transport pathway. The results showed that the filopodial actin filament is a sensory system for EGF detection. In addition, HT29 cells may use the filopodial actin filament to distinguish the presence or absence of the chemoattractant EGF. Furthermore, we demonstrated the high selectivity of PD153035 for EGFR and the reversibility of binding to EGFR. We anticipate that the proposed single-cell method could be applied to construct a rapid screening method for the detection and therapeutic evaluation of many types of cancer during chemotaxis.

**Keywords:** chemotaxis; epidermal growth factor (EGF); epidermal growth factor receptor (EGFR); tyrosine kinase inhibitor; PD153035; optical tweezers; single-cell platform

## 1. Introduction

Cancer cell chemotaxis in the surrounding microenvironment is an essential component of tumor progression and metastasis [1]. Chemotaxis has been recognized as a target for therapeutic intervention, a therapeutic endpoint, and a prognostic marker [2]. The chemotactic response of cancer cells consists of three major steps: chemosensing, polarization, and locomotion. Cancer cells respond to gradients of chemotactic factors by increasing the nucleation and polymerization of the actin filaments [3], where the polymerization of actin leads to directional migration of cancer cells in response to chemoattractant gradients. Recently, receptor tyrosine kinases such as epidermal growth factor receptor (EGFR) have been reported as signaling factors that mediate chemotactic responses in EGF/EGFR signaling [1,2]. Therefore, a rigorous understanding of the mechanism of cancer cell chemotaxis will help us develop novel concepts and strategies for cancer therapy.

Activation of EGFR controls intracellular signaling pathways that regulate cell proliferation, apoptosis, angiogenesis, adhesion, motility, and invasion [4–6]. In colorectal cancer, glioma, head and neck cancer, and non-small cell lung cancer (NSCLC), EGFR expression or high expression is recognized as a particularly promising therapeutic target for anticancer therapies [7–12]. Therefore, different EGFR inhibition strategies have been reported, such as the use of monoclonal antibodies, recombinant EGF vaccines, antisense oligonucleotides, and small-molecule EGFR tyrosine kinase inhibitors (EGFR-TKIs) [13].

Inhibition of protein tyrosine kinase activity by adenosine triphosphate (ATP)-site-directed compounds has become an attractive target for rational drug design in the last few years [14]. Quinazoline derivatives such as Iressa (*gefitinib*) and Tarceva (*erlotinib*) have been widely used for targeted therapies (reviewed in [15]; see also [16,17]). In Japan, Iressa is the most frequently used second-line single-agent targeted therapy; outside Asia, the most frequently used single-agent targeted therapy is Tarceva (reviewed in [18]).

Numerous in vitro studies have shown that Iressa and Tarceva inhibit the EGFR tyrosine kinase by 50%, i.e., $IC_{50}$, at concentrations of 27–33 nM [19,20] and 20 nM [21,22], respectively. Another quinazoline-type compound, 4-*N*-(3′-bromo-phenyl)amino-6,7-dimethoxyquinazoline (PD153035), has been reported to be not only a highly selective reversible inhibitor of EGFR ($IC_{50}$ = 29 pM) [23,24] but also a deoxyribonucleic acid (DNA) intercalator [25–27]. Nowadays, due to ongoing advances in various nanomaterials and nanotechnologies, advanced approaches are required to probe the real-time cellular responses to PD153035. These advances motivated us to examine and address the aforementioned pharmacological issues regarding multitargeted TKI PD153035 in more detail.

In our previous study, we presented a single-molecule approach using optical tweezers to determine the binding mode and binding affinity constant of PD153035 to DNA for the first time. We found that PD153035, although this drug exhibits a noticeable increment in contour length of DNA molecules in 1 mM sodium cacodylate, has weak DNA-binding ability at physiological salt concentrations (100 mM for a monovalent salt) [28]. In addition, we proposed a single-cell approach using a biofunctionalized quantum dot (QD) to demonstrate retrograde transport of EGF–EGFR complexes along filopodia to the A431 cell body [29]; this finding supports the idea that filopodia may serve as a sensory system for the cell that assists in regulating signaling [30]. Previous studies also demonstrated that an EGF-coated bead can induce leading-edge actin protrusion for cells overexpressing the EGFR [1,31]. Here, we applied high-resolution optical tweezers and microfluidic systems with accurate sample temperature control to actively conduct spatial and temporal regulation of cell locomotion, in order to simulate the chemosensing process mediated by EGF/EGFR signaling (Figure 1). In addition, we applied beads with and without an EGF coating positioned close to the filopodia to stimulate HT29 cells overexpressing the EGFR locally. Therefore, we can further apply this platform to quantify the effect of PD153035 on the EGF–EGFR transport pathway and validate whether PD153035 is a specific and reversible inhibitor of the EGFR tyrosine kinase at the single-cell level.

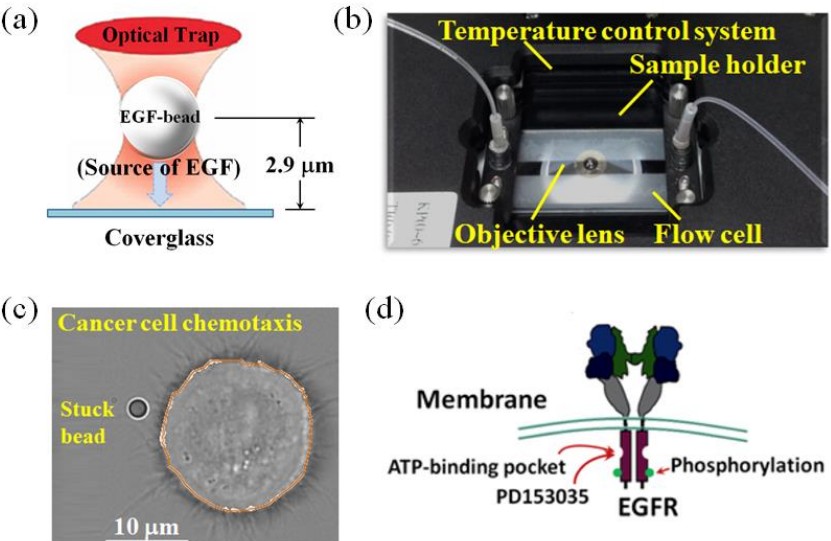

**Figure 1.** Schematic representation of the high-resolution optical tweezers applied to conduct spatial and temporal regulation of cell locomotion to simulate the chemosensing process mediated by EGF/EGFR signaling. (**a**) The trapped bead positioned close to the filopodia is stuck onto the cover glass surface via optical tweezers. (**b**) A single-cell detection platform, including temperature control system and flow delivery system, is integrated into an inverted fluorescence microscope. (**c**) A typical bright-field image of individual cellular responses during stuck-EGF-bead stimulation, where the stuck EGF-coated bead is intended to contact the filopodial actin filament of the HT29 cell. The corresponding 2D cell body contour is also shown, highlighted with an orange line. (**d**) A schematic representation of PD153035 binding to the active ATP catalytic site of the intracellular domain of EGFR.

## 2. Materials and Methods

### 2.1. Experimental System

To conduct spatial and temporal regulation of chemotaxis, the EGF-coated bead was manipulated by our high-resolution optical tweezer system; details of this system have been described previously [28]. Our optical tweezer system was integrated into an inverted microscope (TE2000U, Nikon, Tokyo, Japan) incorporating a Nd:YAG laser (1064 nm, VA-II-N-1064, Beijing, China) for trapping. The laser beam was expanded five-fold with lens pairs to slightly overfill the back aperture of the objective lens (Plan Apo 60x/1.40 oil, Nikon, Tokyo, Japan), and directed into the objective lens via a dichroic mirror (FF720-SDi01-25×36, Semrock Inc., New York, NY, USA).

The advantage of the near-infrared laser (1064 nm) is that both water and biological specimens are comparatively transparent to it, as such specimens have a low absorption coefficient in the near-infrared region [32], which implies that both thermal damage caused by the trapping laser and thermal convection inside the flow chamber can be neglected. Here, we applied optical tweezers to manipulate the trapped bead close to the filopodia, and the trapped bead was then moved downward via optical tweezers and finally stuck onto the cover glass surface (Figure 1a). Note that when the trapped bead was attached to the surface of the collagen-1-coated cover glass, we then turned off the trapping laser. Hence, the image acquisitions during EGF-bead stimulation could be performed within the same focal plane without the trapping laser. Therefore, the EGF bead could move in both the lateral and z-direction only when filopodial actin fiber touched the EGF-coated bead. This approach minimized the optical damage caused by the trapping laser and allowed us to perform local stimulation of HT29 cells overexpressing EGFR for long-term observation of the mode of cell locomotion during EGF-coated bead stimulation and for quantifying the effect of PD153035 on the EGF–EGFR transport pathway.

Our previous study showed that the bead could be captured when the laser power ranged from 5 to 15 mW [33]. However, in this study, we applied optical tweezers to manipulate the EGF-coated bead close to the filopodia (lateral movement), and the bead was then moved downward via optical tweezers in the z-direction and finally attached to the collagen-1-coated cover glass. The laser power inside the flow chamber was maintained at 50 mW to manipulate the trapped bead stably in three dimensions, and this allowed us to conduct the procedure mentioned above within one minute.

We designed a temperature-controlled microfluidic flow chamber system (Figure 1b) incorporated into an optical tweezer system to probe the real-time cellular behavior in response to the chemoattractant EGF at the single-cell level, where the flow chamber was assembled by placing one layer of parafilm (100 μm thick), cut in the chamber shape, between a microscope glass slide and a collagen-1-coated cover glass (170 μm thick).

To provide precise temperature control of the microfluidic flow chamber under the microscope, four electric resistance rod elements were embedded symmetrically within the aluminum heating plate, where the bottom of the flow chamber was in contact with the heating plate sample holder, such that the temperature inside the flow chamber was stable and uniform. Note that the desired temperature of the heating plate could be set with the PID temperature controller; it was possible to adjust the temperature to within ±0.1 °C of the desired temperature. Detailed information can be found in our previous paper [33]. addition, an automated syringe attached to the flow chamber was used for buffer exchange, delivering various solutions, i.e., quantum dots, Alexa Fluor$^®$ 635 phalloidin, and PD153035, and streptavidin-coated bead conjugation to EGF (EGF-coated bead), in an optimally controlled manner.

Figure 1c shows a typical bright-field image of individual cellular responses during stuck-EGF-bead stimulation, where the stuck EGF-coated bead was intended to contact the filopodial actin filament of the HT29 cell. To ensure consistency of experimental conditions, the position of the EGF-coated bead must be within the range that the filopodial actin fiber can touch and must be about a particle size away from the edge of the cell membrane, in order to observe the mode of locomotion during EGF-coated bead stimulation and quantify the effect of PD153035 on the EGF–EGFR transport pathway. The corresponding two-dimensional (2D) cell body contour is also shown in Figure 1c, highlighted with an orange line. Figure 1d shows a schematic representation of PD153035 binding to the active ATP catalytic site of the intracellular domain of EGFR, where PD153035 is an extremely potent EGFR inhibitor that competes with ATP for the intracellular catalytic site of the EGFR, thereby inhibiting the tyrosine kinase activity of the EGFR.

### 2.2. Bead Conjugation to Epidermal Growth Factor

We applied streptavidin-coated bead conjugation to biotin EGF as a point source of the chemoattractant, instead of producing a growth factor gradient by diffusion, to locally stimulate HT29 cells. To this end, an optically trapped bead was coated with the chemoattractant EGF and positioned close to the filopodia of the HT29 cell to conduct the spatial and temporal regulation of cell locomotion to test whether the chemosensing is directly mediated by EGF/EGFR signaling. Here, we labeled 0.4 pM of streptavidin-coated beads (1.87 μm, Bangs Laboratories, Fishers, IN, USA) with 6.36 μM of biotin EGF (Molecular Probes, Invitrogen, Carlsbad, CA, USA) at a ratio of $1:1.6 \times 10^7$, where an excess of biotin EGF was used not only to produce uniformly coated beads but also to saturate the surface of the streptavidin-coated beads. Note that the affinity of biotin for streptavidin is exceptionally high, and therefore the biotin–streptavidin interaction is essentially irreversible.

### 2.3. Cell Culture and Reagents

In this study, the human colon cancer cell line HT29 overexpressing the EGFR was used as research model to investigate cancer cell chemotaxis and response to TKI PD153035 at the single-cell level. The HT29 cell line was purchased from the Food Industry Research

and Development Institute (FIRDI) (Hsinchu, Taiwan) and cultured in DMEM supplemented with 10% fetal bovine serum (FBS), 100 IU/mL of penicillin, and 100 mg/mL of streptomycin (Gibco, Grand Island, NY, USA) under standard culture conditions (37 °C, 95% humidified air and 5% $CO_2$). HT29 cells were trypsinized, counted and resuspended at the appropriate densities ($\approx 2 \times 10^5$ cells/mL) in fresh culture medium. The dissociated HT29 cells were then introduced into the inlet port of the flow chamber using an automated syringe pump at a constant volumetric flow rate of 0.5 μL/sec. During the experiments, the temperature inside the flow chamber was 37 °C.

To distinguish the relationship between the EGF–EGFR complex and the actin cytoskeleton, both Alexa Fluor® 635 phalloidin (Invitrogen Cat. no. A34054) and quantum dots conjugated to epidermal growth factor (EGF, Molecular Probes) were used. The Alexa Fluor® 633 phalloidin molecule, a high-affinity F-actin probe, is optimal for fixed and permeabilized cultured cells. Therefore, HT29 cells were fixed in 4% paraformaldehyde (PFA), followed by further incubation with Alexa Fluor® 635 phalloidin for 5 min at 37 °C. HT29 cells were also incubated with 10 nM of QD–EGF at 4 °C to target the epidermal growth factor receptor (EGFR) and to identify the relationship between the EGF–EGFR complex and the actin cytoskeleton. To inhibit the EGFR tyrosine kinase activity, HT29 cells were incubated with medium containing 1 μM of PD53035 for 1 h at 37 °C. Note that treatment of HT29 cells with various solutions was also automatically conducted using the flow delivery system.

### 2.4. Epi-Fluorescence Imaging

Fluorescence images of cells were obtained with an inverted microscope (TE2000U, Nikon) equipped with an objective lens (Plan Apo 60×/1.40 oil, Nikon, Tokyo, Japan) with a 1.5× additional magnification lens, band-pass filters for QD 565 and Alexa 635, and an EMCCD camera (Luca EM DL658M, Andor, CT, USA). These fluorescent images were acquired with a temporal resolution of 50 ms, and the pixel size at the sample plane was 110 nm.

### 2.5. Image Analysis and Single-Cell Movement (Velocity and Trajectory) Model

To probe the cellular response and visualize cell locomotion of HT29 cells during the process of chemosensing, the EMCCD camera was used to acquire images at a frame rate of 1 fps for imaging of cell locomotion.

The previous study demonstrated that the EGF-coated magnetic bead could induce leading-edge actin protrusion, i.e., localized activation of the EGFR-induced localized actin polymerization [1]. Here, we applied optical tweezers to manipulate the EGF-coated bead to actively conduct spatial and temporal regulation of cell locomotion to test whether the leading-edge actin protrusion would move toward the EGF-coated bead due to the presence of the chemoattractant.

To quantify cancer cell chemotaxis localized close to the stuck bead and the corresponding response to TKI PD153035 at the single-cell level, we defined the chemosensing axis by extending from the center of the stuck bead to the closest point to the edge of the cell (leading edge) and extending to the distal part of the cell region (trailing edge), as illustrated in Figure 2. Along the chemosensing axis (grey dotted line), the position coordinates of the leading and trailing edges of the cell body contours at time points $t_0$ and $t_1$ are A and B, and C and D, respectively. These position coordinates can be determined using ImageJ software.

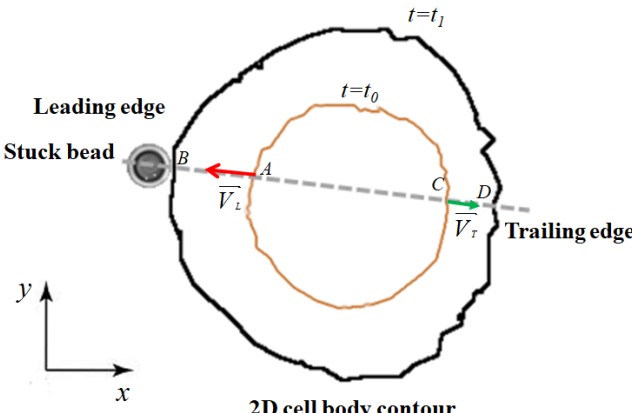

**Figure 2.** Cell movement (velocity and trajectory) model of HT29 cell stimulated with EGF-coated and non-EGF-coated bead stuck onto the cover glass surface. The grey dotted line extending from the center of the stuck bead to the closest point to the edge of the cell (leading edge) and extending to the distal part of the cell region (trailing edge) is defined as the chemosensing axis. The 2D cell body contours are also plotted at time points $t_0$ and $t_1$, highlighted with an orange and a black line, respectively. The ratio of velocity magnitudes, $|\vec{V_L}|/|\vec{V_T}|$, was applied to quantify cancer cell chemotaxis and response to TKI PD153035 at the single-cell level, where $\vec{V_L}$ and $\vec{V_T}$ represent the propagation speeds at the leading and trailing edges.

Here, the 2D cell body contours at time points $t_0$ and $t_1$ are highlighted with an orange and a black line, respectively. Therefore, the propagation speed at the leading and trailing edges of the cell along the chemosensing axis can be determined as follows: $\vec{V_L} = \vec{AB}/(t_1 - t_0)$ (red arrow) and $\vec{V_T} = \vec{CD}/(t_1 - t_0)$ (green arrow). We applied the ratio of velocity magnitudes, $|\vec{V_L}|/|\vec{V_T}|$, to address the cancer cell chemotaxis and response to TKI PD153035 at the single-cell level. Hence, the effect of the chemoattractant EGF on HT29 cell mobility could be obtained and identified.

## 3. Results and Discussion

### 3.1. Relationship between EGF–EGFR Complex and Actin Cytoskeleton

We first investigated the distribution of actin filaments in HT29 cells, where F-actin was visualized using Alexa Fluor® 635 phalloidin (Invitrogen Cat. no. A34054). Here HT29 cells were fixed in 4% paraformaldehyde (PFA), followed by incubation with Alexa Fluor® 635 phalloidin at 37 °C for 30 min. We then compared the bright-field image of the HT29 cell with the fluorescent image of actin filaments; note that both images were taken in the same focal plane for direct comparison (Figure 3a,b). As can be seen, actin filaments were observed in the lamellipodia, colocalized perfectly with the filopodia regions.

We further applied QD–EGF to map the spatial distribution of the EGFR. HT29 cells were incubated with 10 nM of QD–EGF for 30 min at 4 °C and fixed in 4% paraformaldehyde (PFA) to eliminate the possibility of EGFR-mediated internalization. We then compared the bright-field images of the HT29 cell (Figure 3c) with the corresponding epifluorescence image of QD–EGF (Figure 3d). As can be seen, a large accumulation of QD–EGF was observed along the periphery of the cell membrane; there was also colocalization between QD–EGF–EGFR complexes and the filopodia regions. Hence, this finding suggests that filopodia may serve as a sensory system for the HT29 cells in regulating the EGF-induced chemotactic response. For this reason, we used the EGF-coated bead positioned close to the filopodia to test whether cell locomotion exists during EGF-coated-bead stimulation.

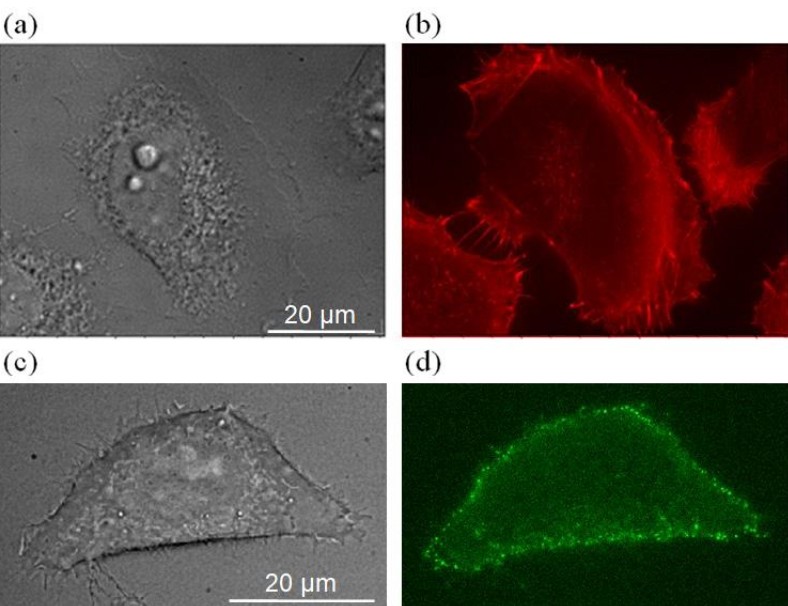

**Figure 3.** Comparison of the bright-field image of HT29 cell (**a**,**c**), the fluorescence image of the distribution of actin filament (**b**), and the simultaneous QD–EGF fluorescence image (**d**).

### 3.2. Mode of Locomotion during EGF-Coated-Bead Stimulation

To conduct spatial and temporal regulation of the HT29 cell to assess the mode of locomotion during EGF-induced chemotaxis, we first delivered the EGF-coated beads into the flow chamber. The EGF-coated bead was manipulated with high-resolution optical tweezers and moved to a position close to the cellular filopodia, and the trapped EGF-coated bead was then stuck onto the cover glass surface with optical tweezers.

Our previous study presented qualitative observations of the leading-edge actin protrusion without quantitative analysis [31]; therefore, some cell images from our previous study will be further scrutinized using the proposed cell movement (velocity and trajectory) model to conduct a quantitative analysis of the mode of locomotion during EGF-coated bead stimulation. We randomly selected three HT29 cells for experimental observation of EGF-coated bead stimulation. Figure 4a,d,g and Figure 4b,e,h) show the bright-field images of the HT29 cell at time points $t_0$ and $t_1$, respectively. According to the proposed cell movement model, the corresponding 2D cell body contours and the value of $|\vec{V_L}|/|\vec{V_T}|$ during stuck EGF-coated-bead stimulation are illustrated in Figure 4c,f,i. The experimental results show that EGF-coated-bead stimulation of HT29 cells causes $|\vec{V_L}| > |\vec{V_T}|$, regardless of the placement position of the EGF-coated bead. This finding implies that the propagation speed at the leading edge of the cell is enhanced due to the presence of EGF-stimulated chemotaxis.

On the other hand, we used the same experimental procedure but changed to a streptavidin-coated bead without conjugation to EGF, as a control to assess whether the migration of HT29 cells towards the stuck bead would occur. As can be seen in Figure 5a,d,g and Figure 5b,e,h, the positions at the leading edge of the HT29 cells at the $t_0$ and $t_1$ time points were nearly the same and were not influenced by the streptavidin-coated bead without conjugation to EGF. In addition, the results show that the streptavidin-coated-bead stimulation of HT29 cells causes $|\vec{V_L}| < |\vec{V_T}|$, regardless of the placement position of the streptavidin-coated bead, as shown in Figure 5c,f,i. This finding implies that the stuck bead in the absence of chemoattractant hinders the propagation speed at the leading edge of the cell, and HT29 cells may use the filopodial actin filament to distinguish the presence or absence of the chemoattractant EGF.

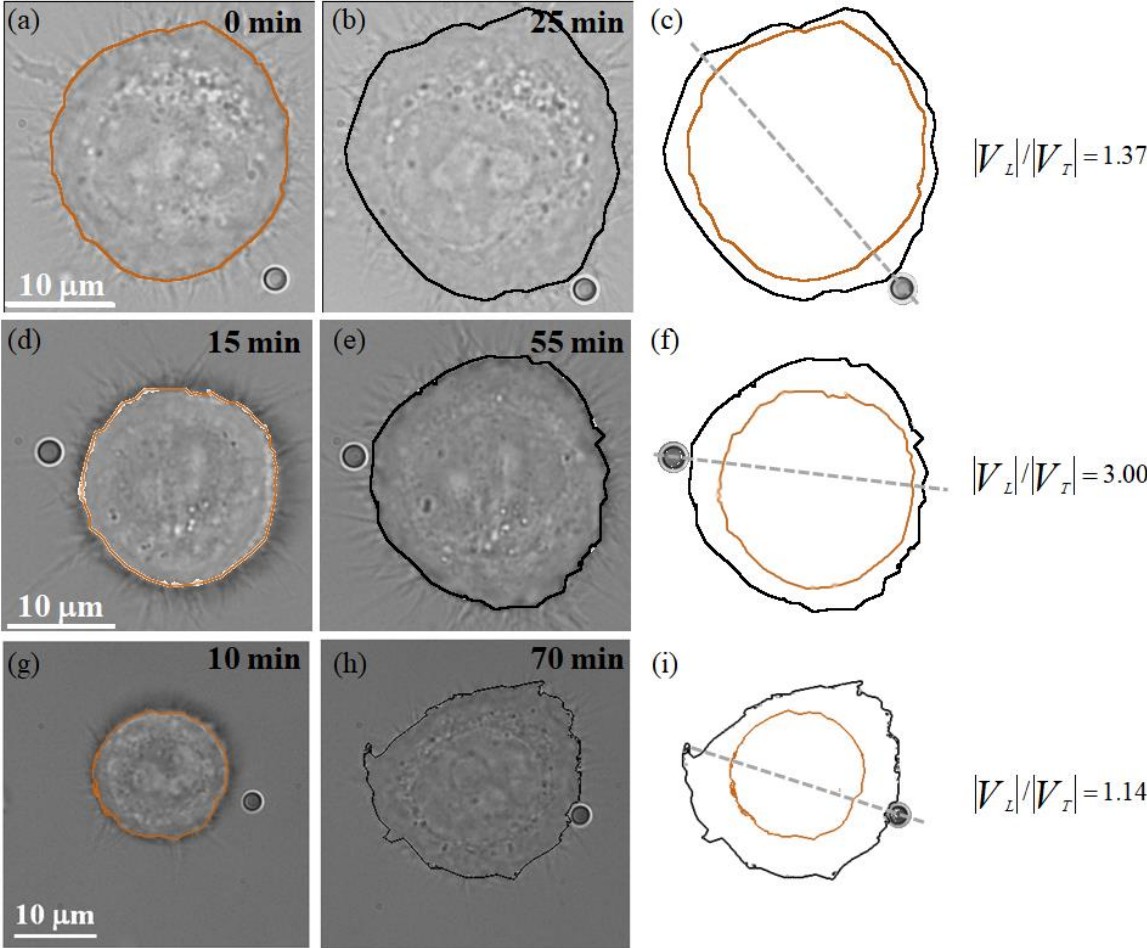

**Figure 4.** Consecutive bright-field images of individual cellular responses during stuck EGF-coated-bead stimulation, where the stuck EGF-coated bead was intended to contact the filopodial actin filament of the HT29 cell. (**a,b,d,e,g,h**) The HT29 cells at the $t_0$ and $t_1$ time points, respectively. (**c,f,i**) The corresponding 2D cell body contours and the value of $|\vec{V_L}|/|\vec{V_T}|$ during stuck EGF-coated-bead stimulation.

In addition to using the value of $|\vec{V_L}|/|\vec{V_T}|$ to address the effect of the stuck bead in the presence or absence of the chemoattractant EGF, we then evaluated the effect of the EGF-coated bead and the streptavidin-coated bead on the propagation speeds at the leading ($|\vec{V_L}|$) and trailing ($|\vec{V_T}|$) edges. The results showed that the propagation speeds of $|\vec{V_L}|$ and $|\vec{V_T}|$ for the EGF-coated bead and the streptavidin-coated bead were $95 \pm 21$ nm/min and $60 \pm 20$ nm/min, and $10 \pm 3$ nm/min and $33 \pm 13$ nm/min, respectively. As can be seen in Figure 6, the value of $|\vec{V_L}|$ for EGF-coated-bead stimulation of HT29 cells is nine times greater than for the case of the streptavidin-coated bead. In addition, the value of $|\vec{V_T}|$ for EGF-coated-bead stimulation of HT29 cells is two times greater than for the case of the streptavidin-coated bead. Note that the stuck bead in the absence of chemoattractant not only hinders the propagation speed at the leading edge of the cell but decreases the propagation speed at the trailing edge of the cell. This finding reveals that the leading edges of the HT29 cells move only when there are advantages to be gained, e.g., in the presence of the chemoattractant EGF; HT29 cells cease to move when there are no advantages to be gained, e.g., in the absence of the chemoattractant EGF.

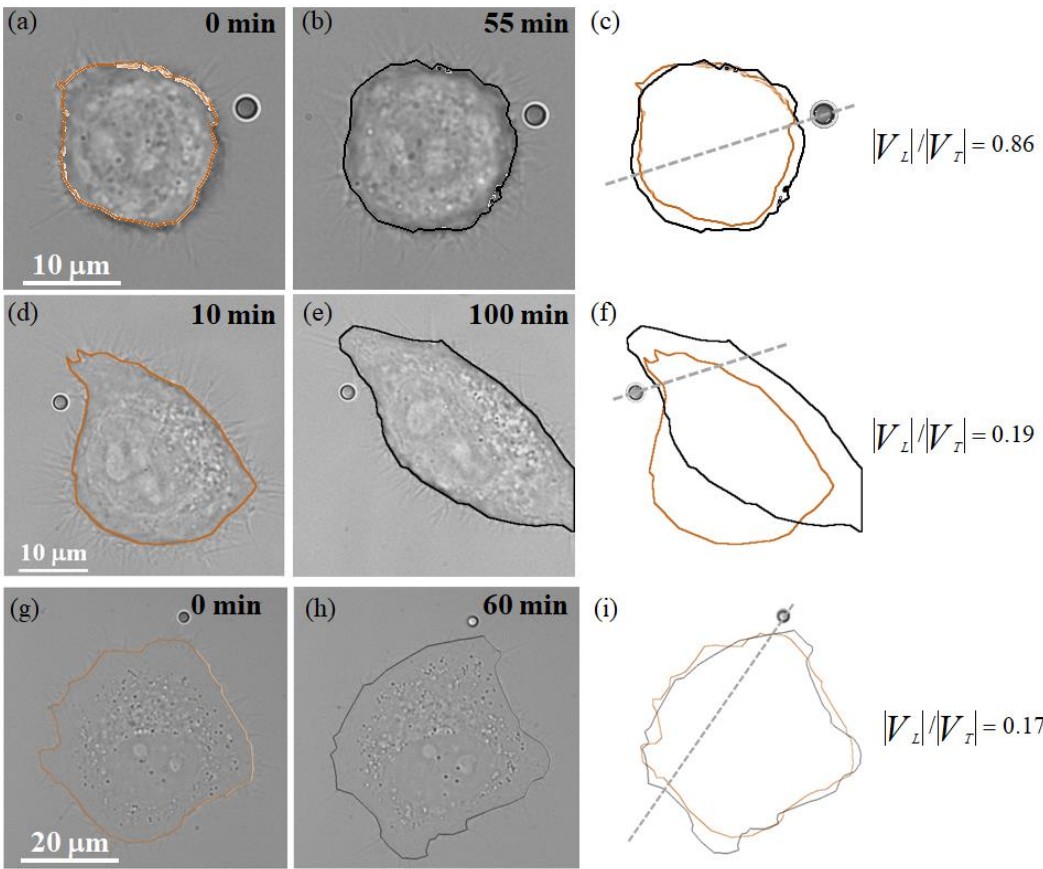

**Figure 5.** Consecutive bright-field images of individual cellular responses during stuck streptavidin-coated-bead stimulation, where the stuck bead was intended to contact the filopodial actin filament of the HT29 cell. (**a**,**b**,**d**,**e**,**g**,**h**) The HT29 cells at the $t_0$ and $t_1$ time points, respectively. (**c**,**f**,**i**) The corresponding 2D cell body contours and the value of $|\vec{V_L}|/|\vec{V_T}|$ during the stuck streptavidin-coated-bead stimulation.

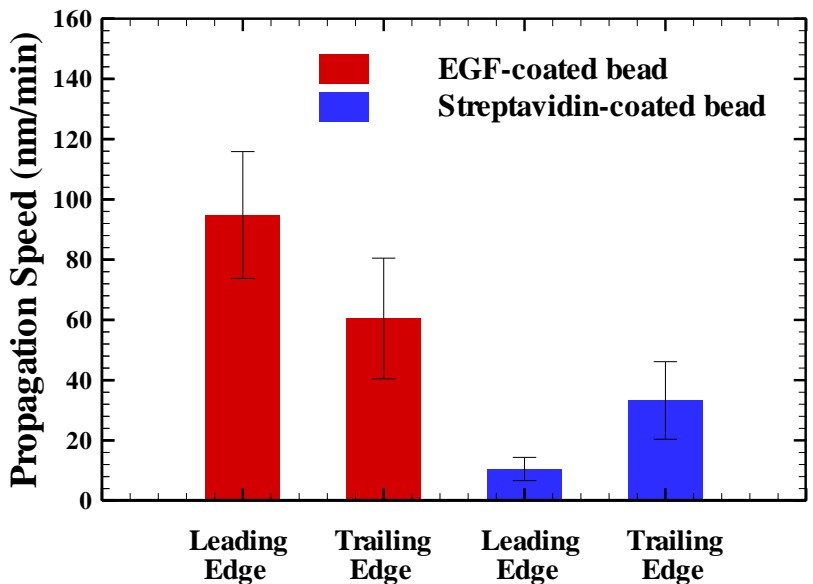

**Figure 6.** The effect of the EGF-coated bead and the streptavidin-coated bead on the propagation speeds at the leading ($|\vec{V_L}|$) and trailing ($|\vec{V_T}|$) edges.

### 3.3. The Effect of PD153035 on the EGF–EGFR Transport Pathway

The previous study demonstrated that PD153035 is a highly selective and reversible inhibitor of EGFR, using an EGFR immunoprecipitation assay [23]. In this study, we further applied the concept of spatial and temporal regulation of HT29 cells using the EGF-coated bead to quantify the effect of PD153035 on the EGF–EGFR transport pathway and to validate the assertion that PD153035 is a specific and reversible inhibitor of the EGFR tyrosine kinase at a single-living-cell level.

The entire single-cell experimental procedure is illustrated in Figure 6. First, HT29 cells were seeded onto a collagen-1-coated cover glass for 1 h. Next, HT29 cells were pretreated with 1 μM of PD153035 for 1 h. Then, in the presence of 1 μM of PD153035, we applied optical tweezers to move the EGF-coated bead into contact with the filamentous actin filaments of the HT29 cell and stuck the EGF-coated bead on the cover glass surface for 1 h. Finally, PD153035 was removed by washing the cells with drug-free medium. During the experiments, the temperature inside the flow chamber was maintained at 37 °C.

Figure 7a–c shows the bright-field images of the HT29 cell in the presence of 1 μM of PD153035 at 0, 30, and 60 min time points, respectively. As can be seen, when the HT29 cell is treated with TKI PD153035, the positions at the leading edge of the HT29 cell over 1h were nearly the same and were not influenced by the EGF-coated bead, which implies that the filopodial actin filament, as a sensory system for EGF detection, could not distinguish the presence of the chemoattractant EGF and demonstrates that PD153035 is a highly selective inhibitor of EGFR. However, after washing the HT29 cells with drug-free medium, it was found that the leading edge of HT29 cells was directly attached to the EGF-coated beads, half an hour after PD153035 removal (Figure 7d–h), which implies that the filopodial actin filament had started to distinguish the presence of the chemoattractant EGF and demonstrates that PD153035 is a reversible inhibitor of EGFR. Finally, at 34 min after PD153035 removal, the EGF-bead was endocytosed into the HT29 cell via an EGFR-mediated pathway, as can be seen in Figure 7i (see also Video S1).

From the above observations, the single-cell approach proposed in this study confirms not only the high selectivity of PD153035 for EGFR but also the reversibility of binding to EGFR, which further demonstrates that the single-cell method could be applied to construct a rapid screening method for the detection and therapeutic evaluation of TKI PD153035.

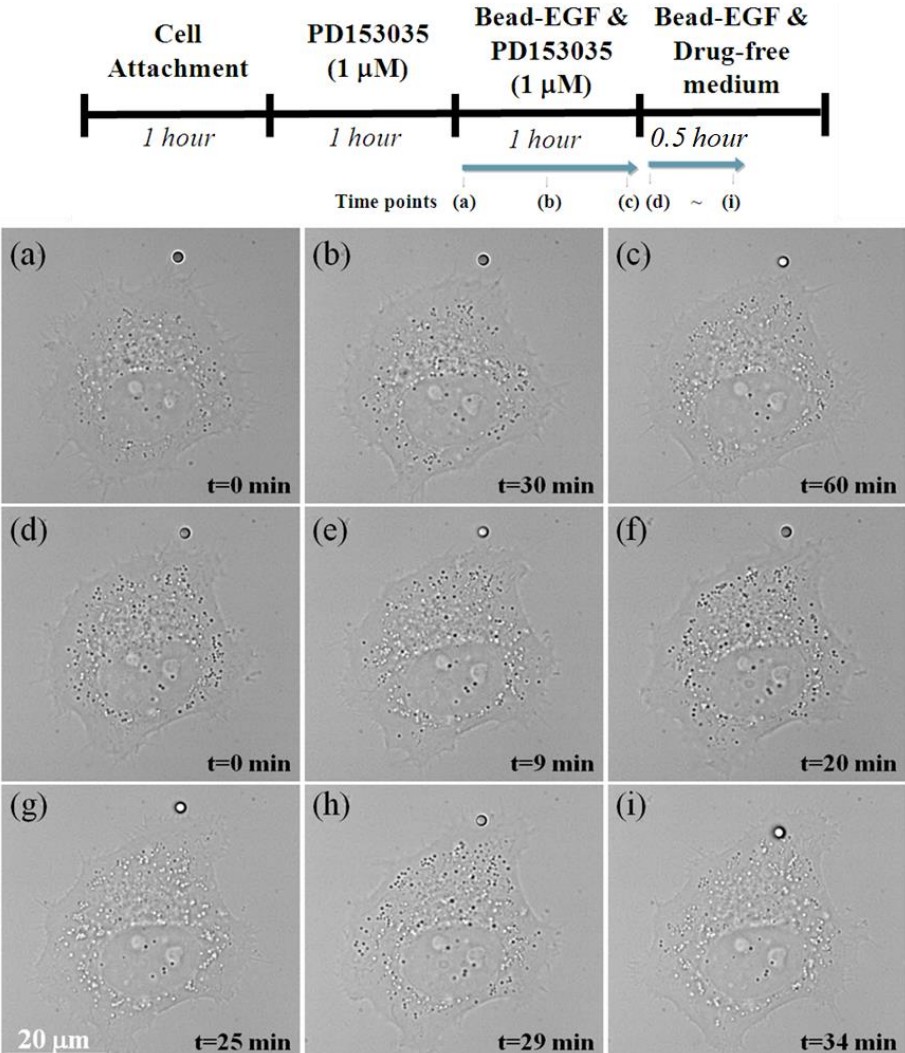

**Figure 7.** Probing the effect of PD153035 on the mode of locomotion during EGF-coated bead stimulation. (**a–c**) The simultaneous bright-field live HT29 cell imaging in the presence of 1 μM of PD153035 for 1 h at 37 °C. (**d–i**) After washing the HT29 cells with drug-free medium, the simultaneous bright-field imaging of the same live HT29 cell was observed at 37 °C within 1 h.

## 4. Conclusions

In this study, we applied an optical tweezer system, together with the platform at the single-cell level, including a temperature control system and flow delivery system, to present an innovative single-cell approach for a chemotaxis assay and for probing the effect of TKI PD153035 on the EGF–EGFR transport pathway. In addition, we proposed a single-cell movement model to quantify the propagation speed at the leading and trailing edges of the cell along the chemosensing axis, and applied the ratio of velocity magnitudes, $|\vec{V_L}|/|\vec{V_T}|$, to address cancer cell chemotaxis. Regarding the relationship between the EGF–EGFR complex and the actin cytoskeleton, there was colocalization between QD–EGF–EGFR complexes and the filopodial actin filaments, which implies that the filopodial actin filament acts as a sensory system for EGF detection. With regard to the spatial and temporal regulation of HT29 cells, the results showed that EGF-coated-bead stimulation of HT29 cells caused $|\vec{V_L}| > |\vec{V_T}|$; however, streptavidin-coated-bead stimulation of HT29 cells caused $|\vec{V_L}| < |\vec{V_T}|$, which implies that the stuck bead in the absence of chemoattractant hinders the propagation speed at the leading edge of the cell, and that HT29 cells may use the filopodial actin filament to distinguish the presence or absence of the chemoattractant

EGF. In addition, the value of $|\vec{V_L}|$ for EGF-coated-bead stimulation was ten times greater than that for the streptavidin-coated bead, and the value of $|\vec{V_T}|$ for EGF-coated-bead stimulation was two times greater than that for the streptavidin-coated bead. Furthermore, we applied the concept of spatial and temporal regulation of HT29 cells using the EGF-coated bead to quantify the characteristics of PD153035 on the EGF–EGFR transport pathway. Observation showed that after washing the HT29 cells with drug-free medium, the filopodial actin filament could start to distinguish the presence of the chemoattractant EGF, which demonstrates not only the high selectivity of PD153035 for EGFR but also the reversibility of binding to EGFR. We anticipate that the proposed single-cell method could be applied to construct a rapid screening method for the detection and therapeutic evaluation of many types of cancer during chemotaxis.

**Supplementary Materials:** The following are available online at https://www.mdpi.com/article/10.3390/photonics8120533/s1, Video S1: External regulation of EGFR-mediated cell motility by EGF-coated beads visualized in real time after PD153035 removal.

**Author Contributions:** Co-first author, J.-C.Y.; conceptualization, T.-S.Y. and J.-C.Y.; methodology, M.M.S.C.; investigation, P.-W.P. and T.-S.Y.; writing—review and editing, T.-S.Y. All authors have read and agreed to the published version of the manuscript.

**Funding:** This research was funded by the Ministry of Education, Taiwan, grant number DP2-110-21121-01-O-06.

**Institutional Review Board Statement:** Not applicable.

**Informed Consent Statement:** Not applicable.

**Data Availability Statement:** Data is contained within the article.

**Acknowledgments:** We acknowledge funding from the Ministry of Education, Taiwan, under grant number DP2-110-21121-01-O-06.

**Conflicts of Interest:** The authors declare no conflict of interest.

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
