# Peer review of "An Optical Tweezers-Based Single-Cell Manipulation and Detection Platform for Probing Real-Time Cancer Cell Chemotaxis and Response to Tyrosine Kinase Inhibitor PD153035"

_photonics, doi:10.3390/photonics8120533_

Round 1
Reviewer 1 Report
Pei-Wen Peng et. al. has studied the response of a single-cell with control temperature in a flow delivery system for chemotaxis assay and the effect of TKI PD153035 on the EGF-EGFR transport in an optical tweezers. The authors are interested in applying their develop technique for the faster screening method for the detection and therapeutic evaluation during chemotaxis.
The overall content is good. This might be interested to the community. But I have following comments that should be addressed by the authors before I recommend the manuscript for the publication in photonics special issue on optical trapping.
- In their model in section 2.5, the choosing of chemotaxis by the author is not well justified. Why the author has choose this axis through the centre of bead and nearest point?
- In continuation of above point, I recommend to evaluate the velocity along the chemotaxis through the bead canter line as well as dissecting the cell area equally almost half in their image for Fig 4 and Fig.5. The author can deduce a bound in the error estimated in their velocity in this process.
- In Fig 6. the washed HT29 bright field image was shown till 34 min only contrary to what they mention in the label for 1 hr. The author should address this issue either by explaining or including more picture till 1 hr.

Reviewer 2 Report
The authors present a single bead-, single cell-based chemotaxis assay to test the effect of drugs on cell movements. First, the authors have shown epifluorescent images of Alexa-phalloidin and EGF-Qdot labeled cells. To claim the colocalization of actin filaments and EGF receptors, it would be required to label the same cell with the two fluorophores.
When testing the mode of locomotion, the authors present three observations with the EGF coated bead-induced chemotaxis and another three observations with the Str coated bead as control. The presented three observations are not sufficient to support their hypothesis. For significant proof, more observations and statistical analysis of the measured variables must be presented.
In these cellular response measurements, the shapes of cells are noticeably different. Once they have compact, spherical initial shape (Fig4g), other times much more flattened, spread shape (Fig4a). Also the time windows of the three measurements are considerably different (25 min, 40 min, and 60 min). The calculated VL/VT ratios also show large variations Th in both groups. The large variability of the experimental parameters, the too few observations presented and the complete lack of statistical analysis does not allow to draw a scientifically significant conclusion.
Minor questions :
When the authors claim that the bead was stuck to the surface, does this mean that the laser was turned off?
How can the spatial resolution claimed in section 2.4 be less than the diffraction limit? For example n case of the 635 nm emission of Alexa with an objective numerical aperture of 1.4 the theoretical limit of resolution equals 276 nm.
Finally, I find the method interesting and of great potential for drug testing, but the authors must provide a better quality of data, therefore I do not recommend the manuscript for publication at this early stage.
Reviewer 3 Report
The authors described the use of optical tweezers to study cancer cell chemotaxis in the presence of EGF coated beads in real time. This technique could serve as a screening method for anticancer agents. The work is well written however I have the following comments,
- How many repeats of each experiment were done? in the manuscript only details of 3 cell measurements for studying the effect of EGF (with and without, Fig 4 and Fig 5), and one cell measurement for probing the effect of PD153035 were discussed. I would recommend to increase the number of measurements to a statistically significant number and also include the standard deviation of the ratio of velocity magnitudes in each case
- The video in supplementary information was described as “External regulation of EGFR-mediated cell motility by EGF-beads visualized in real-time after PD153035 removal.” Was the position of bead moved during this process, it seems like the position of bead at first and last frame (360th) is different. I recommend to clearly explain the video to avoid confusion
Author Response
General Comment: The authors described the use of optical tweezers to study cancer cell chemotaxis in the presence of EGF coated beads in real time. This technique could serve as a screening method for anticancer agents. The work is well written however I have the following comments.
General Response: The authors would like to thank Reviewer #3 for the careful review of our manuscript and for providing us with constructive comments and suggestions to improve the quality of the manuscript.
------------------------------------------------------------------------------------------------------------------------
Comments from Reviewer #3:
- The video in supplementary information was described as “External regulation of EGFR-mediated cell motility by EGF-beads visualized in real-time after PD153035 removal.” Was the position of bead moved during this process, it seems like the position of bead at first and last frame (360th) is different. I recommend to clearly explain the video to avoid confusion.
Author’s Response: Thank you.
Reviewer #3 mentions that the bead is not stuck and sometimes moves during this process.
Indeed, the EGF-bead attached to the collagen-1-coated coverglass is not stuck in a static position due to the viscoelastic properties of the collagen-1-coated gels. However, the purpose is to make sure the EGF-bead will not escape from the attached surface's original position after turning off the trapping laser, and the EGF-bead is moved in both lateral and z-direction only when filopodial actin fiber touches the EGF-bead.

Reviewer 4 Report
In overall, the manuscript is well conceived and well written. The authors have shown a good and practical example of how optical forces can be used to manipulate small particles/cells and demonstrated interesting aspects such as high selectivity of PD153035 for EGFR and the reversibility of binding to EGFR. The manuscript should be considered for publication after addressing a few minor concerns, mentioned below:
- Make sure that all acronyms are clearly explained at their first appearance in the text, for non-expert readers (HT29 cells, ATP, DNA, …)
- There are places where simulate and simulation are used, please, verify if they didn’t take place of stimulate and stimulation, unintentionally.
- In Figures, the grey dotted line used to estimate the cell movement seems very arbitrary, and any slight change would significantly affect the VL/VT ratios (especially in Fig.5), would you comment on this point. I wish there could be a more systematic way to make this measurement.
- What was the laser power used in these experiments and why that value? Could you comment on the choice of the wavelength and the tradeoff between optical forces (laser power) and the risk of cell damage? And what would be the situation in the absence of the cooling flow?
- Does the focused laser beam hold in place both the bead and cell, or only the bead? What is the estimated waist beam size? How do you make sure the trapping forces are not effecting the growth? (any comparison of growth rates with and without the laser, even in literature …)
- Some statements needs a better clarification/formulation for a non-expert reader:
- The trapped bead positioned close to the filopodia was then stuck on the coverglass surface via optical tweezers (Figure. 1(a)) to minimize optically induced damage caused by optical trapping laser.
- Double check for some minor typos and punctuation.
- Missing bracket @ line 56
- …, Fishers, IN, USA) @ line 150
Reviewer 5 Report
Please find attached my comments

Author Response
General Comment: In this manuscript, the authors presented an approach to address chemotaxis and to investigate the response to tyrosine kinase inhibitor at the single-cell level. The authors have shown that the filopodial actin filament can be a sensor for EGF detection. Although the idea of detection of chemoattractant EGF, as the main part of this study, is interesting, I believe at this stage, to prove this claim more discussions should be added in the manuscript. However, the following needs to be addressed before being accepted for publication:
General Response: The authors would like to thank Reviewer #4 for the careful review of our manuscript and for providing us with constructive comments and suggestions to improve the quality of the manuscript. Your concern has been carefully addressed, as can be seen below. Modifications are highlighted by red color words in the markedly revised manuscript to help the Reviewer and the Editor for an easier checking on the changes we made.
------------------------------------------------------------------------------------------------------------------------
Comments from Reviewer #4:
- The authors achieved to stick the EGF-bead using optical tweezers. From the video, it seems that the bead is not stuck and sometimes is moving at z-direction leading to the unfocused image? Thus, I am wondering which is the way that the author stuck the bead on the glass substrate, did they melt the particle on glass substrate? Can the authors clarify this point?
Author’s Response: Thank you.
Reviewer #4 mentions that the bead is not stuck and sometimes moves in z-direction leading to the unfocused image. However, in this study, the image acquisitions are taken within the same focal plane during EGF-bead stimulation, as shown from bright-field HT29 cell images in Fig. 6.
We find that the collagen-1-coated coverglass can facilitate cell attachment and promote both streptavidin-coated bead and EGF-coated bead attached to the surface of the collagen-1-coated coverglass, rather than melting the coated bead on the glass substrate.
Indeed, the EGF-bead attached to the collagen-1-coated coverglass is not stuck in a static position due to the viscoelastic properties of the collagen-1-coated gels. However, the purpose is to make sure the EGF-bead will not escape from the attached surface's original position after turning off the trapping laser, and the EGF-bead is moved in both lateral and z-direction only when filopodial actin fiber touches the EGF-bead.
- The laser power was 50 mW. This is exceptionally high laser power to trap and move a particle of 1.87 μm diameter. Usually, for this size of the particle, i.e. ~2 μm, a maximum of 2 mW is required. Thus, I am wondering how this laser power affects the biotin EGF of the EGF-bead? Maybe affects EGF function. Could the authors clarify this point?
Author’s Response: Thank you.
Indeed, our previous study shows that the bead can be captured when the laser power ranges from 5 to 15 mW [1]. However, in this study, we apply optical tweezers to manipulate the EGF-bead close to the filopodia (lateral movement), and the EGF-bead is then moved downward via optical tweezers in the z-direction and finally attached to the collagen-1-coated coverglass. To complete the aforementioned procedure within one minute, the laser power inside the flow chamber was maintained at 50 mW to manipulate the trapped bead stably in three dimensions.
On the other hand, to minimize the optical damage caused by the trapping laser, the near-infrared laser (1064 nm) is utilized for trapping, where the trapping laser with a wavelength in the near-infrared region is comparatively transparent to biological specimens and such specimens have a low absorption coefficient in the near-infrared region [2].
[1] Cheng, C.M., Chang, M.C., Chang, Y.F., Wang, W.T., Hsu, C.T., Tsai, J.S., Liu, C.Y., Wu, C.M., Ou, K.L., Yang, T.S. Optical tweezers-assisted cross-correlation analysis for a non-intrusive fluid temperature measurement in microdomains Jpn. J. Appl. Phys., 51 (2012), pp. 067002-067005.
[2] Ashkin, A., Dziedzic, J. & Yamane, T. Optical trapping and manipulation of single cells using infrared laser beams. Nature 330, (1987) 769–771.
- The authors also mentioned that use a temperature-controlled microfluidic flow cell system in order to maintain the temperature into a chamber stable at 37° The chamber thickness was 100 μm. This size of thickness generates thermal currents that influence the trapping performance. The reviewer believes that the unfocused image of the bead in the video is due to thermal effects that arise from the chamber thickness and the high laser power. Could the author clarify how they maintain a stable temperature in the chamber? And how do they face the thermal convention that they possibly have during the experimental process?
Author’s Response: Thank you.
Reviewer #4 mentions that the unfocused image of the bead in the video is due to thermal effects that arise from the chamber thickness and the high laser power.
As we have described in Point #2, the near-infrared laser (1064 nm) is comparatively transparent to both water and biological specimens, and such specimens have a low absorption coefficient in the near-infrared region [2], which implied both the thermal damage caused by the trapping laser and the thermal convection inside the flow chamber could be neglected. Note that when the trapped bead is attached to the surface of the collagen-1-coated coverglass, we then turn off the trapping laser. Hence, the image acquisitions during EGF-bead stimulation can be taken within the same focal plane without the trapping laser. Therefore, the EGF-bead can move in both lateral and z-direction only when filopodial actin fiber touches the EGF-bead.
To provide precise temperature control of microfluidic flow chamber under the microscope, four electric resistance rod elements were embedded symmetrically within the aluminum heating plate, where the bottom of the flow cell is in contact with the heating plate sample holder, such that the temperature inside the flow cell is stable and uniform. Note that the desired temperature of the heating plate can be set with the PID temperature controller; it is possible to adjust the temperature within ±0.1 °C of the desired temperature. The detailed information can be found in our previous study [1].
[1] Cheng, C.M., Chang, M.C., Chang, Y.F., Wang, W.T., Hsu, C.T., Tsai, J.S., Liu, C.Y., Wu, C.M., Ou, K.L., Yang, T.S. Optical tweezers-assisted cross-correlation analysis for a non-intrusive fluid temperature measurement in microdomains Jpn. J. Appl. Phys., 51 (2012), pp. 067002-067005.
- How close did the authors position the EGF-bead to HT29 cells? Did they keep this distance stable for each experiment? Did they investigate how this distance influences the response of HT29 cells? Could the authors clarify this point, please?
Author’s Response: Thank you.
Reviewer #4 points out an important issue regarding how close the EGF-bead is to HT29 cell.
To conduct spatial and temporal regulation of HT29 cells to assess the mode of locomotion during EGF-induced chemotaxis, the EGF-bead is first moved to a position close to the cellular filopodia and make sure the filopodial actin fiber can touch the EGF-bead. On the other hand, we randomly select ten individual filopodial actin fibers of the HT29 cell and measure the corresponding mean ± standard deviation fiber length is 4.01±0.28 mm, nearly twice the diameter of the coated bead (1.87 μm).
To ensure the consistency of experimental conditions, the position of the EGF-bead must be within the range that the filopodial actin fiber can touch and must be about a particle size away from the edge of the cell membrane, please refer to figure below (scale bar is 20 mm) and Fig. 1(c) in the revised manuscript, to observe the mode of locomotion during EGF-bead stimulation and quantify the effect of PD153035 on the EGF-EGFR transport pathway.
- Could the authors mention the size of the quantum dot-EGF that used?
Author’s Response: Thank you.
Since the sizes of the Qdot® streptavidin conjugate and biotin-EGF are @15–20 nm [3] and @4 nm [4], respectively, in this respect, QD-EGF @19-24 nm.
[3] https://www.thermofisher.com/order/catalog/product/Q10133MP
[4] Peckys, D., Baudoin, JP., Eder, M. et al. Epidermal growth factor receptor subunit locations determined in hydrated cells with environmental scanning electron microscopy. Sci Rep 3, 2626 (2013).
- The authors estimated the propagation speed, VL and VT, for EGF-bead and streptavidin-bead. They also compare the speed magnitude of each bead. They conclude that for EGF-bead the VL>VT while in case of streptavidin-bead the VL<VT. Which is the relationship of the propagation speed, VT, between EGF-bead and streptavidin-bead, i.e. VT EGF ><~ VT streptavidin? Could the authors comment on this point? Could also provide any conclusion about the sensitivity of EGF in this case?
Author’s Response: Thank you for bringing us this valuable information.
The present results show that filopodial actin filament is a sensory system for EGF detection. In addition, HT29 cells may use filopodial actin filament to distinguish the presence or absence of the chemoattractant EGF, where the leadingedge actin protrusion would move toward the EGF-bead due to the presence of chemoattractant.
We find that the value of for EGF-bead stimulation of HT29 cells is ten times larger than that for the case of streptavidin-coated bead. In addition, the value of for EGF-bead stimulation of HT29 cells is two times larger than that for the case of the streptavidin-coated bead. Note that the stuck bead in the absence of chemoattractant EGF not only hinders the propagation speed at the leading edge of the cell but decreases the propagation speed at the trailing edge of the cell.
This finding reveals that the leading edge of the HT29 cells move only when there are advantages to be gained, e.g., in the presence of chemoattractant EGF; HT29 cells cease when there are no advantages to be gained, e.g., in the absence of chemoattractant EGF.
(Please refer to the first paragraph on page 8 and Figure 6 on page 9 in the markedly revised manuscript).
- Minor Comments: Abstract: “This study focused on three prespectives including simulation of the chemosensing process…” I don’t understand the term “simulation”, There is no theoretical model that simulate the experimental process. Did the authors means monitoring? Stimulating? Could the authors clarify this point? Figure 5 (a) and (b). The size of the bead in (a) seems larger than this in (g). Maybe you need to modify the size bar.
Author’s Response: Thank you.
We have modified the wording and the size bar accordingly. Please refer to the abstract on page 1 and Figure 5 on page 9 in the markedly revised

Round 2
Reviewer 2 Report
In their response the authors claim: " We found a significant difference in the distance between A and B when HT29 cells were stimulated with EGF-coated and non-EGF-coated bead the positions between A and B of the HT29 cells were different and were influenced by EGF-bead; however, the positions between A and B were nearly the same and were not influenced by the streptavidin-coated bead without conjugation to EGF."
I am attaching a file with the statistical analysis of the measured data. The result of all the t-test show that the observations are not significant supposing alpha=5%. Three observation are way too few to claim any of the conclusions about the attractant or repellent effect of the tested beads.
It is also noticable that only one cell out of the three shows a relevant increase in its VL/VT ratio. (3.00 versus 1.37 and 1.14)
The velocity trajectory model also must be improved because the displacement of the cell boundary is way more complex than just describe it by the movement of two points. For example in Fig5f the gray line is with an angle to the supposed velocity vector of the trailing edge.
To conclude I think that these results are preliminary and may suggest a potential significance, but that must be proved through increasing the number of observations.

Author Response
General Comment: Three observations are way too few to claim any of the conclusions about the attractant or repellent effect of the tested beads.
General Response: The authors would like to thank Reviewer #2 for the careful review of our manuscript and for providing us with constructive comments and suggestions to improve the quality of the manuscript.
------------------------------------------------------------------------------------------------------------------------
Comments from Reviewer #2:
- I am attaching a file with the statistical analysis of the measured data. The result of all the t-test show that the observations are not significant supposing alpha=5%.
Author’s Response: Thank you for your effort and bringing us this valuable information.
We agree that values of are not statistically sensitive to distinguish the effect of the stuck bead in the presence or absence of the chemoattractant EGF; therefore, we try to separate these two propagation speeds, and to evaluate the effect of the EGF-coated bead and the streptavidin-coated bead on the propagation speeds at the leading () and trailing () edges. The results show that the propagation speeds of and for both the EGF-coated bead and the streptavidin-coated bead are 95±21 (nm/min) and 60±20 (nm/min) and 10±3 (nm/min) and 33±13 (nm/min), respectively. As it can be seen in Figure 6, the value of for EGF-bead stimulation of HT29 cells is nine times greater than that for the case of streptavidin-coated bead. In addition, the value of for EGF-bead stimulation of HT29 cells is two times greater than that for the case of streptavidin-coated bead. Note that the stuck bead in the absence of chemoattractant not only hinders the propagation speed at the leading edge of the cell but decreases the propagation speed at the trailing edge of the cell. This finding reveals that the leading edge of the HT29 cells move only when there are advantages to be gained, e.g., in the presence of chemoattractant EGF; HT29 cells cease when there are no advantages to be gained, e.g., in the absence of chemoattractant EGF.

Reviewer 3 Report
Thanks you for updating the manuscript
- Assuming the propagation speed was calculated from the optical micrographs, it make more sense to express them in micrometers, rather than nm, if its too small may be express them as um/hour.
- Please include the number of measurements performed in each case, for example “the streptavidin-coated bead are 95±21 (nm/min) 353 and 60±20 (nm/min) and 10±3 (nm/min) and 33±13 (nm/min), respectively (based on x independent measurements)”
- Again, assuming more than one measurement was performed, add a sentence about the reproducibility of experiment described in Figure 7 in the main text (I would recommend to add additional data as supplementary information)
Round 3
Reviewer 2 Report
The new analysis shown on Fig. 6. does not provide leap in statistical significance. Supposing that the +- values are SD, the 95% confidence intervals for the difference in the velocities of the leading and trailing edges gives -18 to 88 for EGF coated beads and -10 to 56 for Str coated beads. As you can see none of them are significantly different from zero. No matter how you rearrange the representation of the data, based on only three observations, statistically significant conclusions cannot be drawn. The only way to reach statistical significance is to increase the number of experimental observations.
Besides the insignificance, the method by which the velocities are determined are not conclusive and well determined. There could be a better method than measuring of the velocity of only one point of the cell's circumference. The three average velocities in case of EGF are calculated from very different time windows: 25 min, 40 min, 60 min. This is not acceptable for comparison. Similarly with Str beads delta t values are 55min, 90 min and 60 min respectively.
With all due respect, I suggest that you reconsider your experimental strategies on this work.